# Effect of Application of Biostimulants on the Biomass, Nitrate, Pigments, and Antioxidants Content in Radish and Turnip Microgreens

**Stefania Toscano** [1], **Daniela Romano** [1,*] and **Cristina Patanè** [2]

1   Department of Agriculture, Food and Environment, Università degli Studi di Catania, 95123 Catania, Italy
2   CNR-Istituto per la BioEconomia (IBE), Sede Secondaria di Catania, Via P. Gaifami 18, 95126 Catania, Italy
*   Correspondence: dromano@unict.it; Tel.: +39-095-4783306

**Abstract:** Microgreens are a functional food that is very appreciated for their good taste and product features. They are produced all year without fertilizers and pesticides. In this paper, the effects of the application of commercial and natural biostimulants on the yield and nutraceutical properties of turnip greens and radish microgreens were investigated. The experiment consisted of four treatments based on biostimulants (Bio-1: TRAINER®; Bio-2: AQUAMIN®; Bio-3: leaf moringa extract; C: distilled water (control)) applied in two species (turnip greens and radish). Fresh and dry biomass, nitrate content, pigments, antioxidants, and antioxidant activity were measured. All biostimulants promoted biomass (both fresh and dry) accumulation in the radish but not in turnip greens. The treatment with biostimulant did not affect plant growth in the radish, while a depressive effect of Bio-1 upon plant growth was observed in turnip greens (−19% smaller than control). In radish, Bio-3 led to microgreens with the highest chlorophyll *a* content (+75% with respect to the control). Bio treatments did not affect the Chl (*a*, *b*, total) content in turnip greens. Biostimulants significantly lowered the nitrate content compared to the control (−27% nitrates) and significantly promoted TPC (+19% over the control) in the radish. They also stimulated antioxidant activity (DPPH), with the highest value in Bio-1, in the turnip, and in Bio-2 and Bio-3, in the radish. Conclusively, biostimulant treatments showed a positive effect on microgreens and, in particular, on those of the radish, improving various nutraceutical parameters.

**Keywords:** *Brassica rapa* L. subsp. *oleifera* (DC.) Metzg; *Raphanus sativus* L. novel food; natural biostimulants; polyphenol content; chlorophylls; total sugars





## 1. Introduction

Microgreens are considered a "functional food" for their beneficial effects on human health [1]. They are appreciated for their good taste and flavour, texture, and leaf coloration. They are young plantlets, usually harvested 7–21 days after sowing when the first pair of true leaves are developed [2]. Microgreens are normally produced all year without fertilizers and pesticides [3]. This product is characterized by a high content of phytonutrients and secondary metabolites [3,4].

Growth conditions have been proven to significantly modulate the qualitative profile of microgreens [5]. Among them, great attention should be paid to substrate material [6], as well as the nutrient addition strategies [5]. In consideration of the short cultivation cycle of microgreens, the reduction in chemical substances that could leave residues is of interest.

A new, innovative, environmentally friendly approach is the application of natural plant biostimulants (PBs) to various crops. The mechanisms activated by biostimulants are still not completely known, and therefore, they are under intensive investigation [7–9]. The definition of biostimulants by new Regulations (EU) 2019/1009 underlines that "the function is to stimulate plant nutrition processes independently of the product's

nutrient content" [10]. This definition emphasizes plant biostimulants as being distinguished by their agricultural functions, and they can be derived from a wide range of raw materials [11].

In the last decade, protein hydrolysates and natural plant extracts, including those of tropical origin, have been widely used as plant biostimulants for their beneficial effects on crop productivity and nutritional efficiency [12]. According to Schaafsma et al. [13], protein hydrolysates are reported as "*mixtures of amino acids, polypeptides, and oligopeptides that are manufactured from animal or plant protein sources using partial hydrolysis*". These compounds are offered in different forms: liquid, soluble powder, and granular, and can be used in seed, foliar, and soil applications [14].

Legume-derived biostimulants containing soluble peptides and free amino acids are easily absorbed by leaves and are translocated to other plant parts in a few hours [15]. These biostimulants are also able to stimulate the absorption and assimilation of N, significantly increasing the productivity of crops [12,16].

Plant extracts are also considered biostimulants if they are able to enhance nutrition efficiency, abiotic stress tolerance, and/or crop quality traits. This category of compounds is being used to replace chemical fertilizers by stimulating plant growth and increasing efficiency in the use of nutrients without having contrasting effects on yields and the quality of various crops [17]. Among the natural biostimulants, we find the *Moringa oleifera* leaf extract (MLE). This species contains various essential nutrients (minerals, fibers, proteins, sugars, free proline, free amino acids, vitamins, and phytohormones, including zeatin, auxins, and gibberellins, as well as many antioxidants) [18–20]. MLE is able to increase seed germination and the first phase of growth in different species [21,22]. A concentration of 4% MLE as a foliar spray on pepper seedlings in the nursery was able to increase the height, fresh and dry weight, number of leaves, and leaf area of pepper seedlings [23]. MLE induces the translocation of reserve food material from the sink (cotyledons or endosperms) to the source (embryo). It leads to an increase in amylase activity which breaks down starch into reducing sugars. Thus, MLE potentially enhances the seed germination percentage, seedling height, biomass, leaf number, and leaf area [20]. The foliar application of natural plant biostimulants can also promote the development of beneficial epiphytic bacteria in plants [24], which enhance the plants' ability to absorb nutrients and self-protect against abiotic stressors and suboptimal conditions [25,26].

The application of biostimulants could be considered a good production strategy for obtaining a high yield of nutritionally valuable vegetables [27]. However, the mechanisms of action for biostimulants are difficult to establish due to the complex pool of bioactive molecules. Indeed, the effects of these compounds are not always positive due to the constituents and the concentration. Furthermore, the effects of biostimulants are different depending on the species because plants can show different thresholds of sensitivity to one or more bioactive molecules [14].

The aim of the paper was to verify the beneficial effects of the application of commercial and natural biostimulants on the yield and nutraceutical properties of turnip greens and radish microgreens. In particular, the effects on fresh and dry biomass, nitrate and sugar contents, chlorophylls, carotenoids, antioxidants, and antioxidant activity, were investigated.

## 2. Materials and Methods

### 2.1. Plant Material and Experimental Conditions

This study was conducted on two species of microgreens: turnip greens (*Brassica rapa* L. subsp. *oleifera* (DC.) Metzg) and radish (*Raphanus sativus* L.) (CN Seeds, Ltd., Pymoor, Ely, Cambridgeshire, UK).

The experiment was carried out February–March 2022 in a cold greenhouse at the Department of Agriculture, Food and Environment (Di3A), University of Catania (Catania, Italy; 37°31′ N 15°04′ E; 20 m above sea level). During the experimental period, the mean temperature was 16.2 °C, and the relative humidity (RH) was 56.4%. The maximum

temperature measured during the experiment was 30.9 °C, and the minimum was 3.5 °C. Seeds of the two species were manually sown in a sowing substrate ('Brill® Semina Bio', Agrochimica S.p.A., Bolzano, Italy) and vermiculite in plastic perforated trays (32 × 22 × 4.5 cm) of a 1 cm depth. Seeds used per tray were approx. 3500 for radish, and approx. 8000 for turnip greens.

The experiment consisted of two species (turnip greens and radish) and four biostimulants (Bio-1, Bio-2, Bio-3, control). Treatments were arranged in a randomized complete block design with three replicates (i.e., trays) per treatment and species.

Irrigation was applied at 2-day intervals to let the seedlings grow under non-limiting water conditions.

Microgreens were harvested 14 days after sowing.

### 2.2. Biostimulants Application

Three different biostimulants were used separately in this experiment: the commercial TRAINER® and AQUAMIN® (Hello Nature Italy SRL, Rivoli Veronese, Italy) (respectively Bio-1 and Bio-2), and leaf extract of moringa (*Moringa oleifera* Lam.) (Bio-3). Bio-1 and Bio-2 are both plant-derived biostimulants obtained by the enzymatic hydrolysis of proteins from the seeds of legumes. As reported by the manufacturer, they contain, respectively, 310 and 620 g kg$^{-1}$ of free amino acids and soluble peptides. The *M. oleifera* leaf extract (LME) was prepared according to Toscano et al. [28]. The leaves of *M. oleifera* were shade-dried, then finely ground with a mill. The powder was mixed in distilled water (50 g in 200 mL). The blend was maintained for 48 h at 25 °C, then filtered through filter paper Whatman No 1 and diluted in water [29]. A Tween-20 (0.05%) wetting agent was added to the spray solutions.

Throughout the experiment, the seedlings of the two species were uniformly sprayed with one of three biostimulants using a 2 L plastic sprayer. The following concentrations were used for commercial biostimulants: 5 mL L$^{-1}$ of TRAINER® (Bio-1) and 2 g L$^{-1}$ of AQUAMIN® (Bio-2). The concentrations used for TRAINER® and AQUAMIN® were those recommended by the manufacturer. For LME (Bio-3), the concentration used was a 1:30 (*v*/*v*) dilution, as indicated by Zulfiqar et al. [29].

The biostimulant solutions were applied to the microgreens 1, 5, and 10 days after seedling emergence for a total of three applications. At each application, the seedlings were sprayed until dripping. Contextually, untreated microgreens of both species were sprayed with distilled water as a control.

### 2.3. Seedling Measurement

At harvest time, microgreens in an area of 4 × 4 cm$^2$ were collected for total fresh (FY) and dry yield (DY); this last was conducted in a thermo-ventilated oven at 70 °C until constant weight. After that, seedling height (cm) and unit fresh weight (FW) was measured on a total of 45 seedlings (15 seedlings randomly selected within each replicate), and seedling dry weight was determined in a thermo-ventilated oven at 70 °C until constant weight for dry matter percentage calculation. Fresh samples of microgreens were instantly frozen in liquid nitrogen and stored at −80 °C until laboratory analysis. All chemical analyses were carried out in triplicate.

### 2.4. Chlorophyll and Carotenoid Pigments Measurement

Chlorophyll *a* and *b* (Chl *a* and Chl *b*) and carotenoid contents were determined according to Lichtenthaler [30]. Briefly, microgreen fresh samples (100 mg) were homogenized in 5 mL of methanol (99%) and incubated in the dark for 24 h at 4 °C. Absorbance was read spectrophotometrically (UV-1900i UV-VIS Spectrophotometer, 230 V, Shimadzu, Tokyo, Japan) at 665.2 nm, 652.4 nm, and 470 nm. The following formulas were used for the quantification:

Chl $a$ = 16.75$A_{665.2}$ − 9.16$A_{652.4}$
Chl $b$ = 34.09$A_{652.4}$ − 15.28$A_{665.2}$

$$\text{Carotenoids} = (1000A_{470} - 1.63\text{Chl }a - 104.96\text{Chl }b)/221$$

### 2.5. Total Sugars Measurement

Total sugars were determined spectrophotometrically according to the Cocetta et al. [31] method. The anthrone reagent (10.3 mM) was prepared by dissolving anthrone in ice-cold 95% $H_2SO_4$. One g of the microgreen fresh sample was homogenized in 3 mL of distilled water and centrifuged at $3000 \times g$ for 15′ at room temperature (RT). After that, 0.5 mL of the extract was mixed with 2.5 mL of anthrone reagent and incubated in ice for 5′; then, it was heated in a water bath at 95 °C for 10′ and left to cool in ice. Absorbance was read spectrophotometrically at 620 nm, and glucose was used as standard (0 to 0.05 mg mL$^{-1}$) ($R^2 = 0.9995$).

### 2.6. Nitrate Concentrations Measurement

Nitrate concentration was determined according to Cataldo et al. [32]. One g of the microgreen fresh sample was homogenized in 3 mL of distilled water and then centrifuged at $4000 \times g$ for 15′. Then, 20 μL of supernatant was added to 80 μL of salicylic acid (5% in sulfuric acid) and to 3 mL of NaOH 1.5 N. Absorbance was read in the spectrophotometer at 410 nm, and $KNO_3$ was used as a standard (0, 1, 2.5, 5, 7.5, and 10 mM $KNO_3$) ($R^2 = 0.9983$).

### 2.7. Ascorbic Acid Measurement

The ascorbic acid content was determined using the protocol of Janghel et al. [33]. Microgreen fresh samples (1 g) were homogenized in 10 mL of 5% oxalic acid and centrifuged at $4000 \times g$ for 5′. One mL of extract was added to 2 mL of methyl viologen (0.1%) and 2 mL of NaOH (2 mol L$^{-1}$). Absorbance was read spectrophotometrically at 600 nm against the radical blank. The concentration of ascorbic acid was expressed using L-ascorbic acid as a standard curve (100–500 μg mL$^{-1}$) ($R^2 = 0.9998$).

### 2.8. Total Phenolics Measurement

For the total phenolic compounds (TPC) measurement, a 1 g sample of fresh weight was extracted in a solution containing acetone and water (1:1). Then, the extracts were vortexed and incubated for 15 h at 20 °C. Next, 100 μL of the supernatant was mixed with 0.5 mL of the Folin–Ciocâlteu reagent (Sigma-Aldrich, Milan, Italy) before 6 mL of distilled water and 1.5 mL of $Na_2CO_3$ (20%) were added, and the samples were incubated at 20 °C for 2 h. Absorbance was read spectrophotometrically at 765 nm [34]. TPC is reported as gallic acid equivalent (GAE, mg 100 g$^{-1}$ FW) ($R^2 = 0.9993$).

### 2.9. Radical Scavenging Activity Measurement

The antioxidant activity was measured by the 2,2-diphenyl-1-picrylhydrazyl (DPPH) assay, as reported by Toscano et al. [34]. About 1 g of the microgreen's fresh weight was mixed with 1.5 mL of 80% methanol solution, incubated in an ultrasonic bath for 30′, and centrifuged at $4500 \times g$ at 5 °C for 10′. Then, 0.01 mL of the supernatant was added to 1.4 mL of 150 μM DPPH (solved in 95% methanol) and incubated in the dark for 30′. Absorbance was read spectrophotometrically at 517 nm, and the values were expressed as Trolox equivalents (μmol TE g$^{-1}$ FW) ($R^2 = 0.9988$).

### 2.10. Statistical Analysis

Data were subjected to a one-way ANOVA within each species, using CoStat 6.311 (CoHort Software, Monterey, CA, USA), and differences among the means were assessed using an LSD post hoc test ($p \leq 0.05$). A principal component analysis (PCA) was also performed on morphological and quality traits using the software Minitab 16, LLC. A Pearson's correlation test was conducted among all parameters, separately per species (SigmaPlot11, Systat Software Inc., San Jose, CA, USA).

## 3. Results

### 3.1. Fresh and Dry Biomass Yield

The total fresh and dry biomass per unit area significantly changed with treatments in both species (Table 1). Within the species, an overall significantly lower fresh yield was produced under Bio-2 in turnip greens ($-20\%$ than all treatments), whilst a $+67\%$ and $45\%$ greater fresh yield than the control corresponded, respectively, to Bio-1 and Bio-3, in radish (Figure 1a,c). The same response was observed in terms of dry yield in turnip greens, whilst in radish, all Bio treatments produced higher yields than the control ($+22\%$, on average) (Figure 1b,d).

**Table 1.** Results of one-way ANOVA for fresh (FY) and dry biomass yield (DY), seedling height (H), seedling fresh weight (FW) and seedling dry matter percentage (DM), in microgreens of turnip greens and radish.

|  | df | MS | F | *p* |
|---|---|---|---|---|
| **Turnip greens** | | | | |
| FY | 3 | 25.4626 | 12.97 | 0.0019 |
| DY | 3 | 7026.77 | 9.57 | 0.0050 |
| H | 3 | 4.5819 | 11.4107 | $\leq$0.001 |
| FW | 3 | 53.829 | 1.3702 | 0.3198 |
| DM | 3 | 0.12438 | 1.8585 | 0.2149 |
| **Radish** | | | | |
| FY | 3 | 15,464.18 | 7.983 | 0.0086 |
| DY | 3 | 27.847 | 4.604 | 0.0374 |
| H | 3 | 0.1851 | 0.6289 | 0.5973 |
| FW | 3 | 5267.36 | 4.2204 | 0.0459 |
| DM | 3 | 1.2215 | 4.885 | 0.0324 |

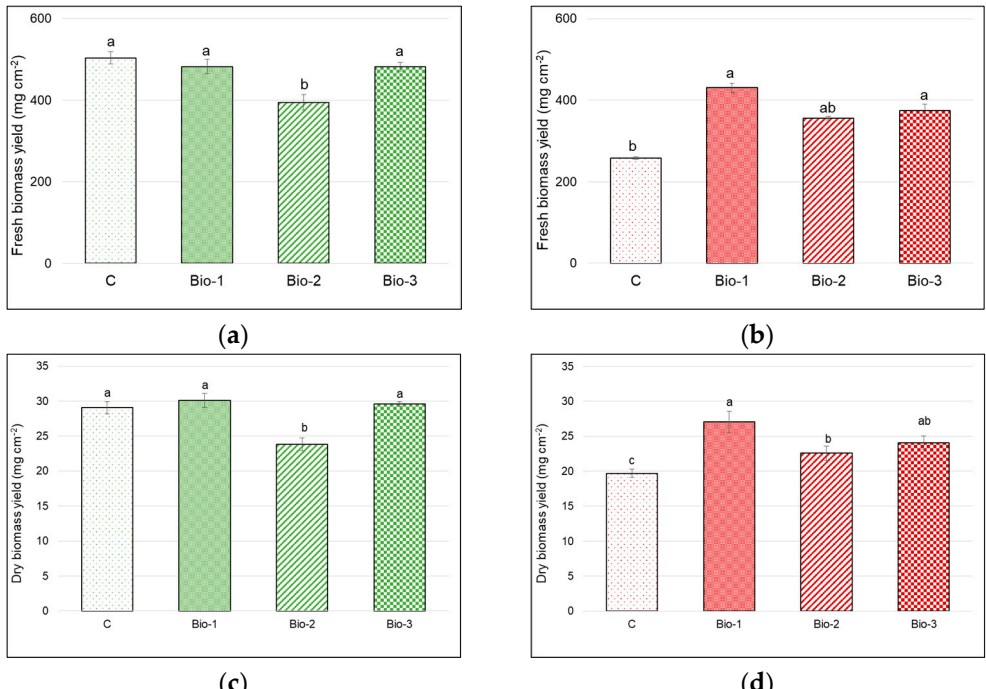

**Figure 1.** Effect of biostimulants (Bio-1 = TRAINER®; Bio-2 = AQUAMIN, and Bio-3 = *Moringa oleifera* leaf extract) on fresh and dry biomass yield (mg cm$^{-2}$) in turnip greens (**a**,**c**) and radish (**b**,**d**) of microgreens. Data are means $\pm$ se (*n* = 3). Different letters indicate significance at $p \leq 0.05$ according to the LSD test.

### 3.2. Seedling Height and Weight

Seedling height was significantly affected by treatments only in turnip greens ($p \leq 0.001$), where Bio-1 gave the overall lowest value (−15%) (Table 1, Figure 2a,b).

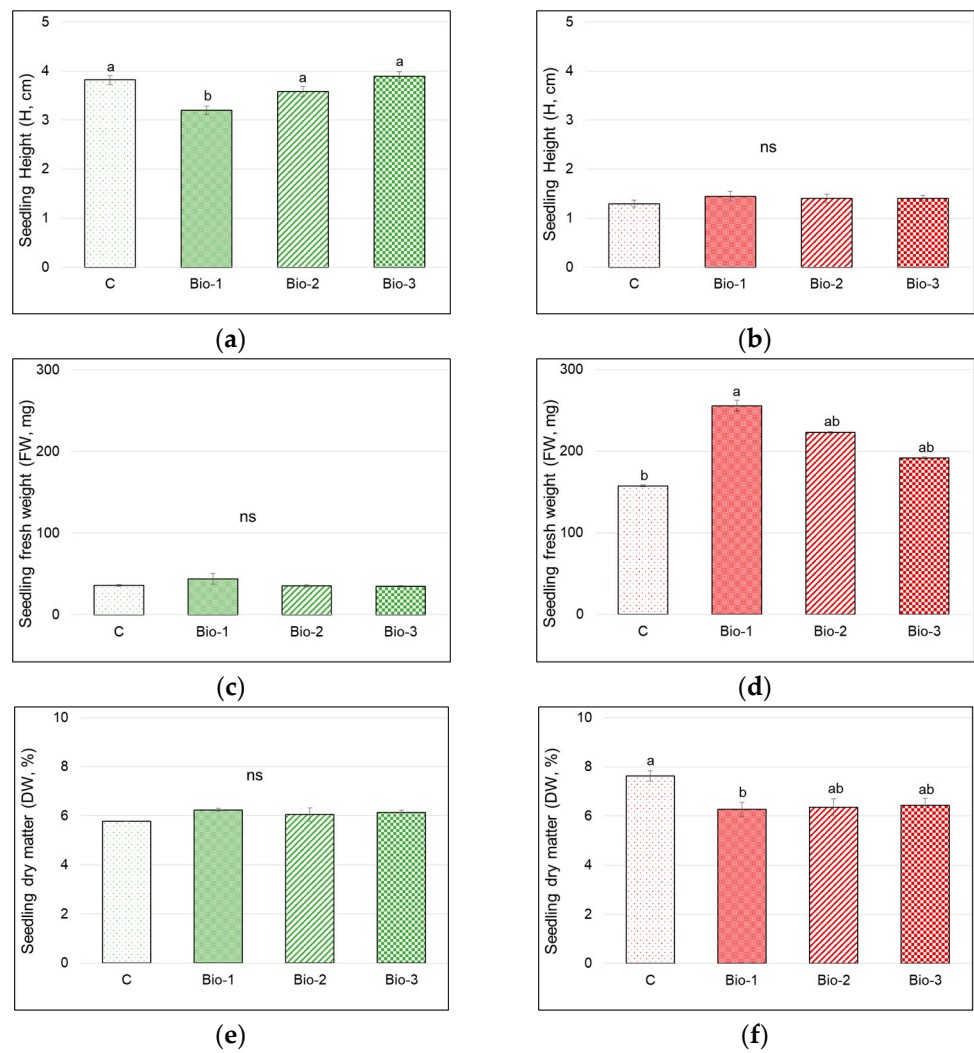

**Figure 2.** Effect of biostimulants (Bio-1 = TRAINER®; Bio-2 = AQUAMIN, and Bio-3 = *Moringa oleifera* leaf extract) on seedling height (cm), seedling fresh weight (mg), and dry weight percentage (%) in turnip greens (**a**,**c**,**e**) and radish (**b**,**d**,**f**) microgreens. Data are means ± se (*n* = 3). Different letters indicate significance at $p \leq 0.05$ according to LSD test.

The fresh weight of single seedlings at harvest was greater in the microgreens of radish (207.2 mg vs. 37.6 mg in turnip greens) (Figure 2c,d). No differences were highlighted in turnip greens among treatments. In the radish, differently from what was observed for height, Bio-1 led to the seedlings having a fresh weight 38% higher than that measured in those untreated ($p \leq 0.05$) (Table 1).

The percent incidence of dry weight on fresh biomass varied with treatments in the radish ($p \leq 0.05$) but not in the turnip ($p = 0.21$). Precisely, in radish, Bio-1 determined a greater water content (i.e., lower DM %) in microgreens compared to the control (Figure 2e,f).

### 3.3. Chlorophyll (a, b, Total) and Carotenoids

The content of Chl *a*, *b*, and the total, significantly changed with treatments in radish only (Table 2). Greater contents of Chl *a* were found in microgreens treated with Bio-1 and Bio-3 ($p \leq 0.001$). For Chl *b* and the total Chl, the highest values corresponded to Bio-3

(respectively + 35–37% than the control) (Table 3). No effects of the biostimulants upon all the chlorophylls were found at ANOVA for turnip greens (Tables 2 and 3).

**Table 2.** Results of one-way ANOVA for chlorophyll *a* (Chl *a*), *b* (Chl *b*), total chlorophyll (Total Chl), carotenoids, chlorophyll *a/b* ratio (Chl *a*/Chl *b*), and chlorophyll/carotenoid ratio (Chl/Car), in microgreens of turnip greens and radish.

| | df | MS | F | p |
|---|---|---|---|---|
| **Turnip greens** | | | | |
| Chl *a* | 3 | 0.00183 | 1.823 | 0.221 |
| Chl *b* | 3 | 0.00074 | 1.326 | 0.332 |
| Total Chl | 3 | 0.00474 | 1.616 | 0.261 |
| Carotenoids | 3 | 0.000053 | 1.900 | 0.208 |
| Chl *a*/Chl *b* | 3 | 0.2063 | 1.287 | 0.3432 |
| Chl/Car | 3 | 3.0432 | 17.909 | ≤0.001 |
| **Radish** | | | | |
| Chl *a* | 3 | 0.01134 | 23.019 | ≤0.001 |
| Chl *b* | 3 | 0.003206 | 12.823 | 0.002 |
| Total Chl | 3 | 0.02651 | 19.600 | ≤0.001 |
| Carotenoids | 3 | 0.000149 | 7.1013 | 0.012 |
| Chl *a*/Chl *b* | 3 | 0.1357 | 2.048 | 0.1857 |
| Chl/Car | 3 | 2.6302 | 16.900 | ≤0.001 |

**Table 3.** Effects of biostimulants (Bio-1 = TRAINER®; Bio-2 = AQUAMIN, and Bio-3 = *Moringa oleifera* leaf extract) on chlorophyll *a* (Chl *a*), *b* (Chl *b*), total chlorophyll (Total Chl), carotenoids (µg mg$^{-1}$ FW), chlorophyll *a/b* ratio (Chl *a*/Chl *b*), and chlorophyll/carotenoid ratio (Chl/Car), in turnip greens and radish microgreens.

| | Chl *a* | Chl *b* | Total Chl | Carotenoids | Chl *a*/Chl *b* | Chl/Car |
|---|---|---|---|---|---|---|
| | (µg mg$^{-1}$ FW) | | | | | |
| **Turnip greens** | | | | | | |
| Control | 0.199 ± 0.022 | 0.084 ± 0.019 | 0.283 ± 0.041 | 0.045 ± 0.002 | 2.484 ± 0.266 | 3.192 ± 0.177a |
| Bio-1 | 0.171 ± 0.002 | 0.055 ± 0.004 | 0.226 ± 0.007 | 0.045 ± 0.001 | 3.116 ± 0.196 | 1.151 ± 0.093b |
| Bio-2 | 0.142 ± 0.015 | 0.050 ± 0.006 | 0.192 ± 0.021 | 0.037 ± 0.004 | 2.885 ± 0.049 | 1.005 ± 0.123b |
| Bio-3 | 0.188 ± 0.025 | 0.071 ± 0.018 | 0.260 ± 0.042 | 0.046 ± 0.004 | 2.783 ± 0.319 | 1.515 ± 0.414a |
| *Significance* | ns | ns | ns | ns | ns | *** |
| **Radish** | | | | | | |
| Control | 0.198 ± 0.008c | 0.072 ± 0.005b | 0.270 ± 0.013b | 0.049 ± 0.002b | 2.761 ± 0.089 | 1.479 ± 0.087b |
| Bio-1 | 0.267 ± 0.017b | 0.099 ± 0.016b | 0.366 ± 0.033b | 0.058 ± 0.002ab | 2.768 ± 0.261 | 2.340 ± 0.375b |
| Bio-2 | 0.246 ± 0.011c | 0.093 ± 0.003b | 0.340 ± 0.013b | 0.055 ± 0.005ab | 2.639 ± 0.110 | 2.097 ± 0.034b |
| Bio-3 | 0.346 ± 0.013a | 0.149 ± 0.006a | 0.495 ± 0.020a | 0.066 ± 0.001a | 2.314 ± 0.005 | 3.699 ± 0.241a |
| *Significance* | *** | ** | *** | ** | ns | *** |

Values (mean ± se) within each column, followed by the same letter, do not significantly differ at $p \le 0.05$ according to the LSD test; ns = not significant; significant at $p \le 0.01$ (**), and 0.001 (***).

As for Chl, carotenoids significantly changed with treatments in the radish ($p \le 0.01$) (Table 2). Among the biostimulants, only Bio-3 had slight but positive effects on this trait over the control (+35%) (Table 3). No significant effect of the biostimulants was ascertained in turnip greens (Tables 2 and 3).

The chlorophyll *a/b* ratio did not show any significant difference in both species ($p > 0.05$) (Tables 2 and 3). The Chl/Car ratio was lower in all biostimulants and the turnip greens ($p \le 0.001$), whilst in radish, Bio-3 treatment led to the significantly highest value (Tables 2 and 3).

### 3.4. Total Sugars and Nitrates Content

Total sugars changed with Bio-treatments in both species (Table 4). Their content was the lowest in microgreens of turnip greens treated with Bio-1 ($-22\%$ than control) ($p \leq 0.05$) (Figure 3a). For the radish, all biostimulants significantly depressed the sugar content with respect to the control ($p \leq 0.001$) (Figure 3b).

**Table 4.** Results of one-way ANOVA for total sugars (TS) and nitrates content, total phenolic content (TPC), ascorbic acid (AscA), and antioxidant activity (DPPH), in microgreens of turnip greens and radishes.

|  | df | MS | F | *p* |
|---|---|---|---|---|
| **Turnip greens** | | | | |
| TS | 3 | 0.00196 | 4.803 | 0.034 |
| Nitrates | 3 | 624.46 | 0.168 | 0.915 |
| TPC | 3 | 330.006 | 6.275 | 0.017 |
| AscA | 3 | 34.015 | 8.1963 | 0.008 |
| DPPH | 3 | 14,671.44 | 19.53 | $\leq 0.001$ |
| **Radish** | | | | |
| TS | 3 | 0.01399 | 72.662 | $\leq 0.001$ |
| Nitrates | 3 | 29,139.92 | 13.53 | 0.0017 |
| TPC | 3 | 771.027 | 6.999 | 0.0126 |
| AscA | 3 | 170.903 | 41.574 | $\leq 0.001$ |
| DPPH | 3 | 37,524.8 | 32.504 | $\leq 0.001$ |

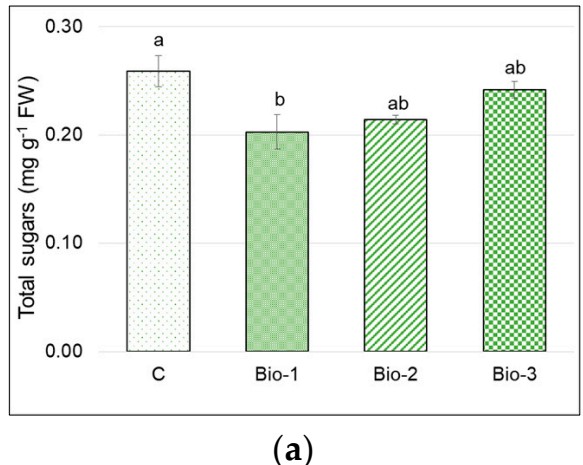 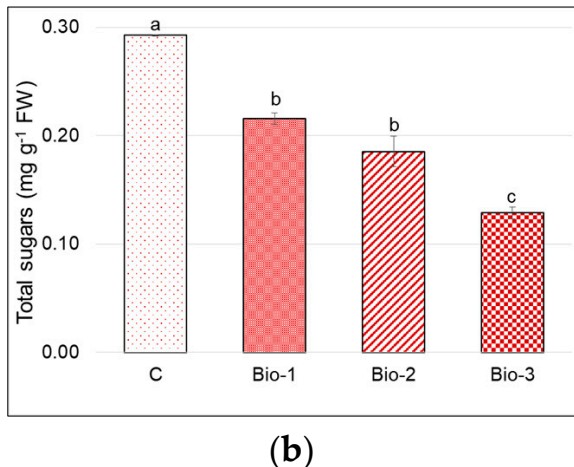

(**a**)                                                                 (**b**)

**Figure 3.** Effect of biostimulants (Bio-1 = TRAINER®; Bio-2 = AQUAMIN, and Bio-3 = *Moringa oleifera* leaf extract) on the total sugars (mg g$^{-1}$ FW) in in turnip greens (**a**) and radish (**b**). Data are means $\pm$ se (*n* = 3). Different letters indicate significance at $p \leq 0.05$ according to the LSD test.

Different from the total sugars, nitrates did not change with treatment in turnip greens (Table 4, Figure 4a). In radishes, all biostimulants led to nitrate contents significantly lower than that of the control ($-27\%$, on average) (Table 4, Figure 4b).

### 3.5. Antioxidants and Antioxidant Activity

The ANOVA results for total phenols, ascorbic acid, and DPPH are reported in Table 4. Between the species, total phenols were $+100\%$ higher in radishes (>159 mg GAE 100 g$^{-1}$ FW). In both species, slight but significant differences were found for TPC ($p \leq 0.05$). In turnip greens, greater contents than the control were measured in Bio-1 ($+32\%$) (Figure 5a). Differently, in the radish, a promoting effect on this trait was exerted by Bio-2 and Bio-3 (Figure 5b).

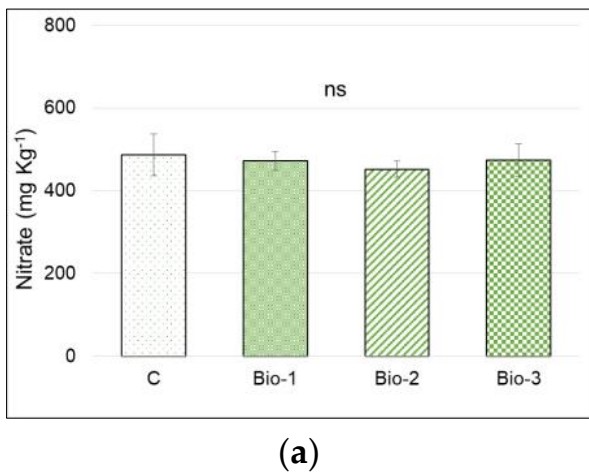
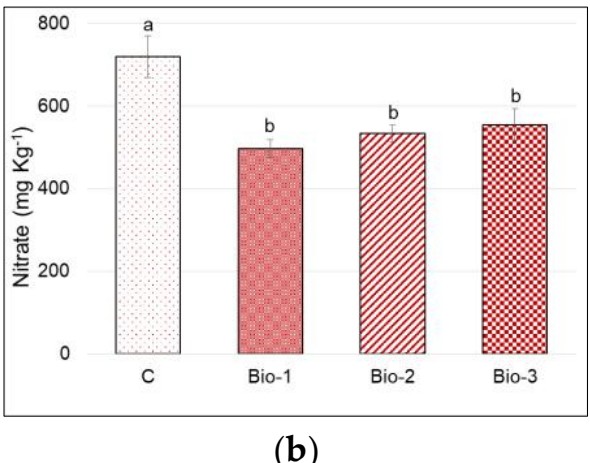

**Figure 4.** Effect of biostimulants (Bio-1 = TRAINER®; Bio-2 = AQUAMIN, and Bio-3 = *Moringa oleifera* leaf extract) on nitrates content (mg kg$^{-1}$) in turnip greens (**a**) and radish (**b**) microgreens. Data are means ± se (*n* = 3). Different letters indicate significance at $p \leq 0.05$ according to the LSD test.

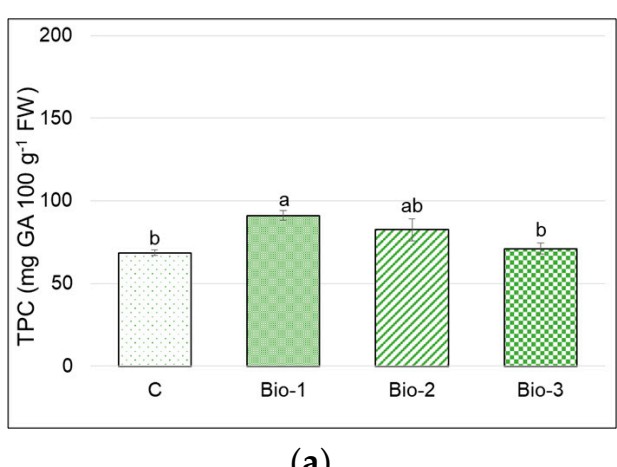
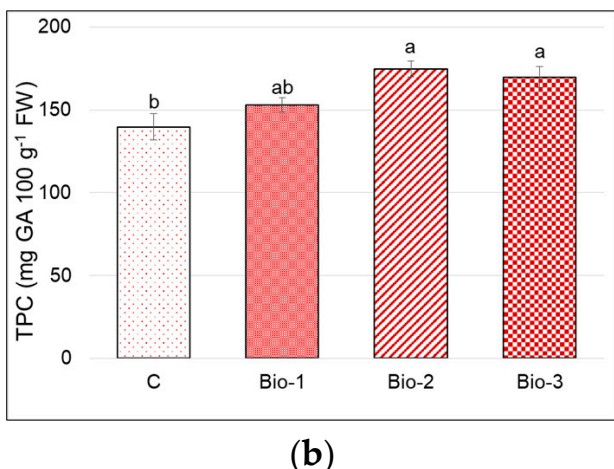

**Figure 5.** Effect of biostimulants (Bio-1 = TRAINER®; Bio-2 = AQUAMIN, and Bio-3 = *Moringa oleifera* leaf extract) on total phenols content (TPC) (mg GA 100 g$^{-1}$ FW) in turnip greens (**a**) and radish (**b**) microgreens. Data are means ± se (*n* = 3). Different letters indicate significance at $p \leq 0.05$ according to the LSD test.

The AscA content is overall similar for the two species (on average, 22.3 and 20.1 mg 100 g$^{-1}$ FW, respectively, in turnip greens and radish) and was significantly affected by treatments (Table 4, Figure 6a,b). However, biostimulants influenced this content differently depending on the species. In turnip greens, AscA in Bio-3 was greater than Bio-1 and Bio-2 but not greater than the control. Contrastingly, in the radish, all biostimulants lowered the AscA content with respect to the control (−48%) (Figure 6b).

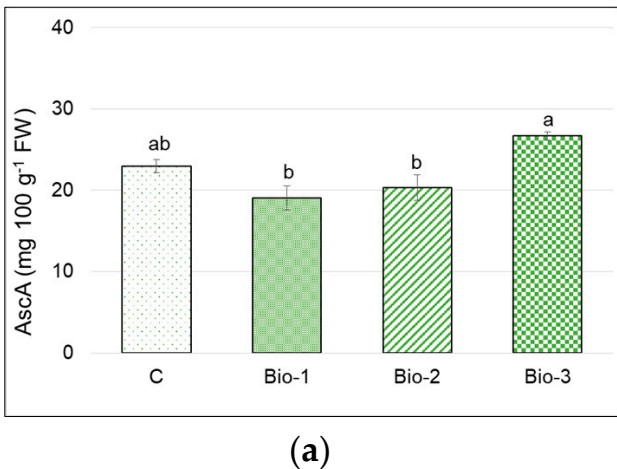
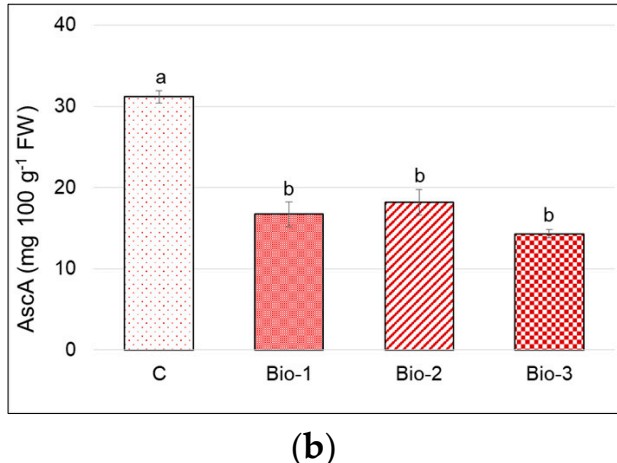

**(a)**                                    **(b)**

**Figure 6.** Effect of biostimulants (Bio-1 = TRAINER®; Bio-2 = AQUAMIN, and Bio-3 = *Moringa oleifera* leaf extract) on the ascorbic acid content (mg 100 g$^{-1}$ FW) in turnip greens (**a**) and radish (**b**) microgreens. Data are means ± se (*n* = 3). Different letters indicate significance at $p \leq 0.05$ according to the LSD test.

According to a higher TPC content, the antioxidant activity expressed as a DPPH free-radical scavenging activity was overall greater in the microgreens of radish (>620 mg TE 100 g$^{-1}$ FW) (Figure 7). In turnip greens, Bio-1 stimulated the antioxidant activity, as revealed by the DPPH values, significantly higher than that of the control (+54%) ($p \leq 0.001$) (Figure 7a). In the radish, both Bio-2 and Bio-3 enhanced the scavenging activity of microgreens over the control (+40% and +33%, respectively) (Figure 7b).

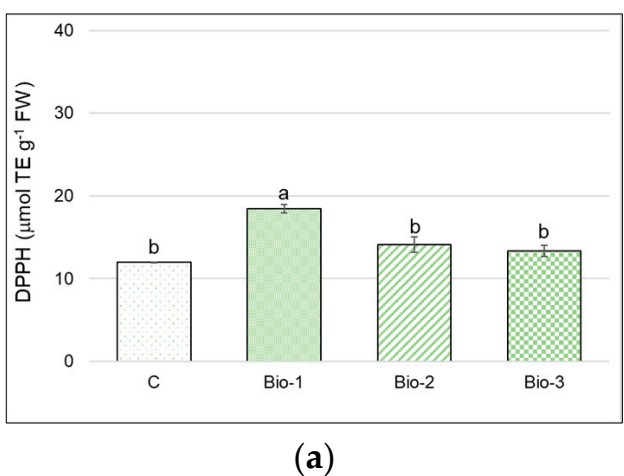
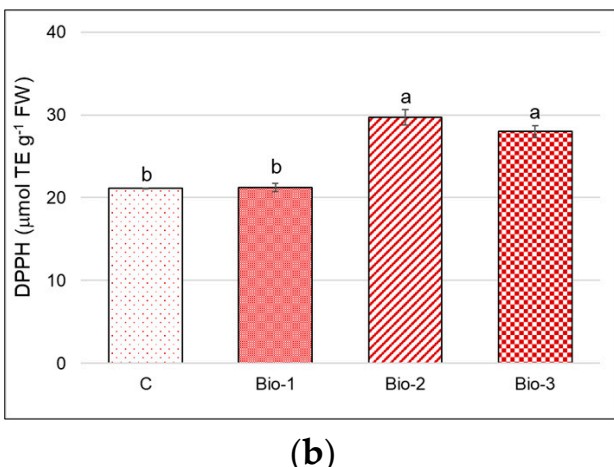

**(a)**                                    **(b)**

**Figure 7.** Effect of biostimulants (Bio-1 = TRAINER®; Bio-2 = AQUAMIN, and Bio-3 = *Moringa oleifera* leaf extract) on the antioxidant activity (DPPH radical scavenging) (μg TE g$^{-1}$ FW) in turnip greens (**a**) and radish (**b**) microgreens. Data are means ± se (*n* = 3). Different letters indicate significance at $p \leq 0.05$ according to the LSD test.

### 3.6. Analysis of Principal Component

PCA was performed for all traits to identify any possible cluster within the analyzed samples. PCA identified three factors with eigenvalue > 1.0 for the turnip greens (PC1, 63.1%, PC2, 28.2%, PC3, 8.7%) and radish (PC1, 64.7%, PC2, 23.0%, PC3, 12.3%). These three factors accounted for a total of 100% of the variance in both species. The first two factors were considered based on their scientific associations and meaningfulness. In turnip greens, the following two groups were identified: (1) T–Bio-1 in the upper quadrant on the left, with DPPH, TPC, unit FW, and DM, and Chl/Car; (2) T–Control and T–Bio-3, in

the lower quadrant on the right include Sugars, AscA, and seedling height. The lower quadrant on the left, with T–Bio-2, clustered with none of the traits considered (Figure 8a). In the radish, the following groups were identified: (1) R-Control, in the upper quadrant on the left, which includes AscA, Unit DM, and Nitrate; (2) R–Bio-2 and R–Bio-3 in the upper quadrant on the right, which includes height, DPPH, TPC, Chl (*a*, *b*, total), Chl/Car, and Car; (3) R–Bio-1 group clustered in the lower quadrant on the right, which includes seedling fresh weight (Unit FW), and fresh and dry biomass (FY and DY) (Figure 8b).

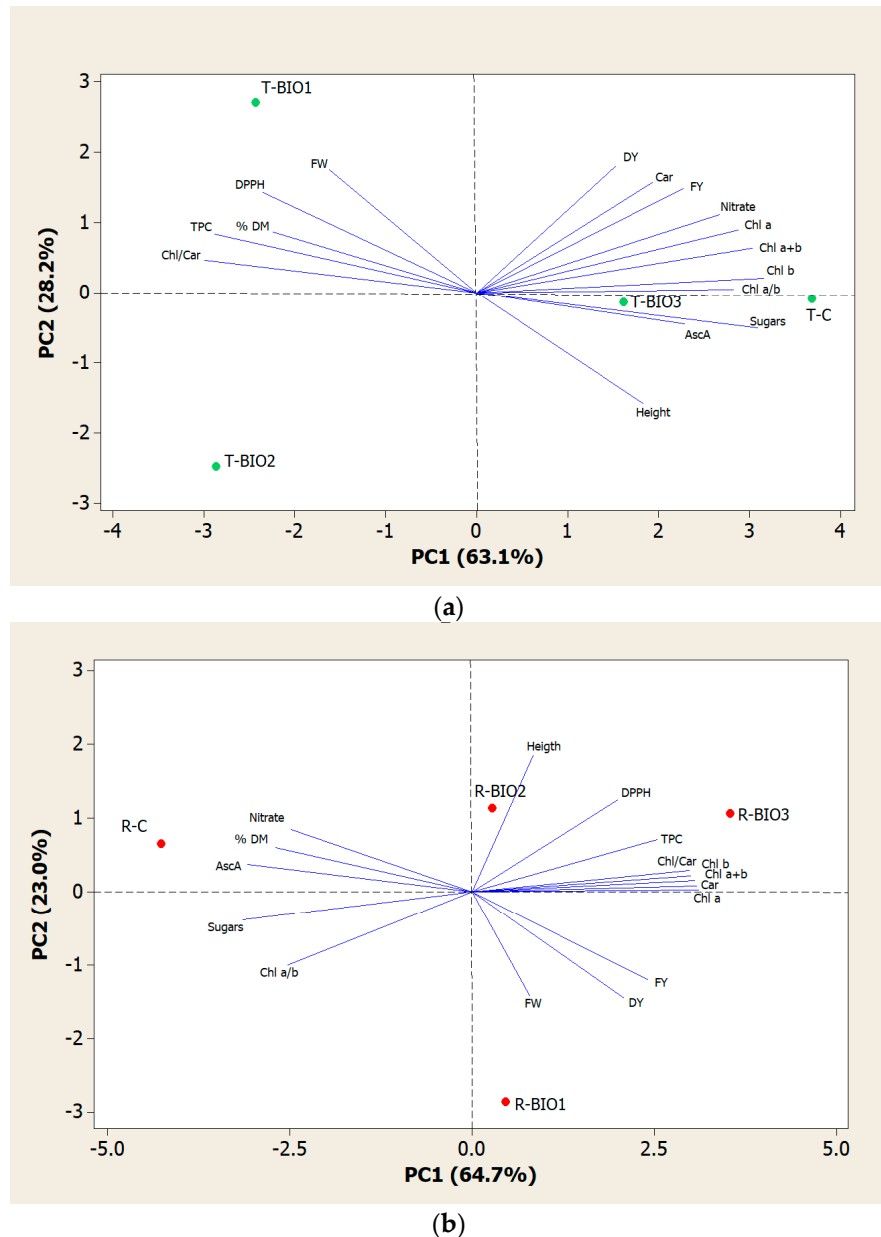

**Figure 8.** Principal component loading plot and scores of PCA for fresh and dry biomass yield (FY and DY), single seedling fresh weight (FW), dry matter percentage (% DM), seedling height, photosynthetic pigments (Chl *a*, Chl *b*, total Chl, carotenoids), total sugars, nitrates content, AscA, TPC, DPPH, for turnip greens (**a**) and radish (**b**) as modulated by treatments with biostimulants.

### 3.7. Pearson's Correlation

The results of Pearson's correlation analysis among all the measured traits are reported separately for each species (Tables S1 and S2). In the microgreens of turnip greens, the fresh and dry yields were positively correlated, respectively, to Chl *a* and carotenoids content

($p \leq 0.05$). Interesting negative correlations of the seedling height were found vs. the dry matter percentage, total phenols, and DPPH. In turn, the total phenols were positively correlated with DPPH ($p \leq 0.05$). These results indicate that in these microgreens, longer seedlings overall had greater moisture content and were poorer in phenol content and, thus, had lower antioxidant activity.

In the microgreens of radishes, among all the significant correlations, both the fresh and dry yield negatively correlated to AscA ($p \leq 0.05$ and 0.01, respectively) and, more, to the nitrate contents ($p \leq 0.001$ in both). Clear positive correlations were also observed for DM% vs. AscA and the nitrate contents (both $p \leq 0.01$). As also found in the microgreens of turnip greens, TPC correlated positively to DPPH ($p \leq 0.01$) and negatively to sugar contents ($p \leq 0.01$).

## 4. Discussion

In this experiment, the effects of the application of three different biostimulants (Bio-1 and Bio-2, based on hydrolyzed proteins from legumes, Bio-3 based on leaf moringa extract) and production and quality aspects, were evaluated in the microgreens of turnip greens and radish grown under cold greenhouse conditions. All quality traits were expressed on a fresh weight basis, considering that microgreens are usually consumed fresh.

Greater fresh and dry biomass yields were produced with the microgreens of turnip. This result may have interesting economic implications, considering that more biomass is produced in the same area unit when adopting turnip greens instead of radishes for microgreen production. However, single microgreens from turnip greens were almost 3-fold longer and weighed more than 4-fold less than those from the radish while having lower dry matter content. As a result, the microgreens of turnip greens were much thinner and more hydrated than those of the radish. Lower dry matter content may indicate an advanced vegetative growth in these microgreens at harvest time, as suggested in the literature [35,36]. However, these results also evidenced a different plant structure developed by the two species during the same time-lapse and under the same conditions of light, temperature, and water supply. This aspect may greatly influence some sensory attributes of microgreens for the two species, e.g., the level of palatability by consumers due to a quite different seedling consistency: fine in turnip greens, fleshy in radish [37].

The results from the present study reveal that biostimulants differed in their effects on the microgreen yield, depending on the species. Indeed, overall, they seemed to stimulate both fresh (up to +67%, with Bio-1) and dry biomass (+22%, on average of Bio-treatments) in radish. However, no effects were observed on the turnip green.

Both Chl *a* and Chl *b* were measured in the microgreens of the two species. Chl *a* is the main pigment involved in photosynthesis; Chl *b* is not necessary for the photosynthetic process; therefore, it is not always present in plant cells performing photosynthesis [36]. However, Chl *b* allows plants to adsorb the light at a wider range of wavelengths, and thus, its increase denotes an adaptation to shade [36]. This aspect is of relevant importance for microgreens grown under a cold greenhouse without artificial light sources, such as those of the present study. Chl *b* was higher in the radish, excluding the control, indicating a possible wider light absorption by this species after Bio-treatment. It is well known that light enhances cell metabolism and the synthesis of secondary metabolites such as phenolics [36].

Carotenoids are non-nutrient bioactive molecules with beneficial effects on human health [38]. Together with chlorophylls, they greatly affect the leaf color of leafy vegetables [39], including microgreens [40], thus contributing to health benefits and visual appearance at the same time [2,36]. Both pigments were present in greater concentrations in the microgreens of radish (total Chl and carotenoids, respectively, +50% and +32% higher than in turnip greens). Higher carotenoids in microgreens than in mature plants were reported in the literature for *Brassica* spp. [41]. As far as the effects of biostimulants are concerned, in radishes, leaf extracts of moringa were effective in increasing Chl and carotenoids by 83% and 35%, respectively, when compared to the control.

Sugars were overall higher in turnip greens, indicating the greater sweetness of these microgreens. The sugar levels found in this study were rather lower than those reported in the literature [42]. Low content of sugars was also found in a study by [43] on microgreens cultivated in different substrates. It is interesting to note that sucrose levels are moderate and lower compared to baby leaves and adult vegetable leaves [44]. Such low sugar concentrations may involve a further shortening of shelf-life (1–2 days) reported by microgreens [45,46]. Low sugar content was also found in a study conducted by Wojdyło et al. [36]. Differently than expected, the effects of biostimulants on this trait were overall depressive. Indeed, they clearly reduced the sugar content in the radish, which was even more than halved in plants treated with moringa.

Tan et al. [47], working on broccoli microgreens, reported a higher sweetness and taste in those produced by local farmers than those that were commercial, associated with higher contents of chlorophyll. However, we found no significant correlation between sugars vs. total Chl in our experiment.

Nitrate contents in microgreens varied with treatment. More than double the content was measured in the radish, indicating that plant species may greatly affect the content of these compounds in microgreens. These results confirm what was found in the literature, where nitrate contents in microgreens ranging from 310 to 1111 mg kg$^{-1}$, depending on species, were reported [5]. Species-specific nitrate content in microgreens was also reported by Toscano et al. [28]. Such variability may be ascribed to a different ability of plant species (even if belonging to the same botanical family) to accumulate these compounds, which, in turn, depends on their photosynthetic and metabolic activity [2,48]. Nitrate content was sensibly reduced in bio-treated seedlings of radish. In this species, all biostimulants were likewise found to be effective in reducing the nitrates of microgreens. These results confirm that, as observed for other components of the produce, the nitrate content in microgreens is influenced by agronomic factors [49]. However, the effect of bio-treatment on this trait was not so clear in turnip greens. Nitrates are an important quality trait in vegetables, and a high intake of them is associated with an increased probability of carcinogenic nitrosamine formation in the stomach [35]. Anyway, the level of nitrates ascertained in the present study was much lower than the maximum levels reported in the imposed European Commission (EC) Regulation No. 1258/2011 (EU) [50].

Microgreens of Brassicaceae are an excellent source of bioactive compounds, such as ascorbic acid (AscA), carotenoids, tocopherols, and phenolic compounds (TP), which contribute to their high nutritional value [51]. In the present study, conducted with the microgreens of turnip greens and radishes, both of the Brassicaceae family, TPC close to (in turnip greens) or much greater (in radish) than 70 mg 100 g$^{-1}$ and ascorbic acid contents higher than 14 mg 100 g$^{-1}$ FW were measured. It has been demonstrated that TPC in microgreens may achieve levels up to 10-fold greater than those in mature plants [52]. However, opposite results were reported by Yadav et al. [3], who found higher TPC and AscA in mature plants of radish compared to their microgreen stage. These last results may be ascribed to a phenolic increase in mature plants in response to a possible stress condition occurring during plant growth [3]. Anyway, microgreens are usually consumed fresh; therefore, both antioxidants, thermolabile vitamin C in particular, are largely retained [53]. The application of biostimulants overall caused reductions in the AscA content of microgreens in both species, with more evident depressive effects on radishes.

Phenolics are an important class of secondary metabolites that are associated with the flavor properties of vegetables, such as color and taste [54]. They also have important bioactive properties, mostly determined by their antioxidant activity [55]. A more complex polyphenol profile than mature plants has been reported in *Brassica* microgreens [56]. In our experiment, the TP content of radishes was more than double in this respect than the turnip greens, confirming how plant species may greatly affect TPC in microgreens [57]. Greater chlorophyll content, Chl *a* in particular, in radishes, may be responsible for its higher TP content [58]. Light enhances cell metabolism and the synthesis of secondary metabolites such as phenolics [36].

Polyphenols content in microgreens has been also reported to change with agricultural practices [59]. In this study, the application of the three biostimulants promoted TPC, with more evident effects of Bio-1 in turnip greens, and of Bio-2 and Bio-3, in the radish. These results demonstrate that the efficacy of bio-treatment in promoting this trait is not unique, but it may greatly change with the type of biostimulant and plant species. TPC usually has an astringent taste, thus influencing the flavor of microgreens [37]. In the present study, a close positive relationship of TPC was observed with the antioxidant activity (DPPH). As a consequence, the antioxidant activity was promoted in the microgreens of turnip greens when treated with Bio-1 and in those of the radish when treated with Bio-2 and Bio-3. However, such a positive relationship was not found for AscA vs. DPPH. Similar results were reported for TPC and AscA vs. antioxidant activity in microgreens from several plant species [53].

PCA analysis was performed to assess the relationships between the morphological and quality traits in the microgreens of the two species. The score plot of all variables generated from the comparison of the first two PCAs revealed three distinct clusters per species, which consisted of different traits depending on the Biotreatment.

## 5. Conclusions

Microgreens are an emerging product category that has been gaining increasing attention over the last decade for their nutritional and organoleptic characteristics.

Our results revealed that the application of biostimulants, based on hydrolised proteins or leaf moringa extract, during growth may influence the final biomass in the microgreens of turnip greens and radishes. Moreover, some parameters, such as chlorophylls, total phenolic content, antioxidant activity, and nitrate content, might be promoted or depressed by biostimulants depending on the species.

Indeed, our results highlight the species-specific efficacy of biostimulants. In fact, more evident responses in the radish to biostimulant application were observed. In this species, Bio-2, based on hydrolised proteins, and Bio-3, based on leaf moringa extract, promoted TPC and DPPH, and all biostimulants reduced the nitrate content significantly.

These effects were not evident in turnip greens, where Bio-2 even reduced the final fresh yield. In this species, only Bio-1, based on hydrolised proteins, enhanced TPC and antioxidant activity, and no effect of the biostimulants on nitrate content was observed.

Further research on biostimulants is needed in order to identify optimal doses and application frequency for each biostimulant, to improve yield, and the nutraceutical traits in microgreens.

**Supplementary Materials:** The following supporting information can be downloaded at: https://www.mdpi.com/article/10.3390/agronomy13010145/s1, Table S1: Correlation matrix with the Pearson coefficient values for all the measured traits in turnip greens microgreens, Table S2: Correlation matrix with the Pearson coefficient values for all the measured traits in radish microgreens.

**Author Contributions:** Conceptualization, D.R. and C.P.; methodology, S.T. and C.P.; software, S.T. and C.P.; validation, D.R.; formal analysis, S.T.; investigation, S.T.; resources, D.R.; data curation, S.T. and C.P.; writing—original draft preparation, S.T., C.P., and D.R.; writing—review and editing S.T., C.P. and D.R.; supervision, C.P. All authors have read and agreed to the published version of the manuscript.

**Funding:** This research received no external funding.

**Institutional Review Board Statement:** Not applicable.

**Informed Consent Statement:** Not applicable.

**Data Availability Statement:** Not applicable.

**Conflicts of Interest:** The authors declare no conflict of interest.

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
