# Peer review of "Effect of Application of Biostimulants on the Biomass, Nitrate, Pigments, and Antioxidants Content in Radish and Turnip Microgreens"

_agronomy, doi:10.3390/agronomy13010145_

Round 1

Reviewer 1 Report

Biostimulants application in Brassica rapa L. subsp. oleifera (D.C.) and Raphanus sativus L. microgreens: effects on biomass, nitrate content, pigments, antioxidants and antioxidants activity

The title is too long and leaves the reader confused, because all the study information was included in the title. the title must be explanatory and the species used do not need to be highlighted as scientific name in the title, but popular name (radish and turnip greens) and put the scientific names in the keyword or in the abstract to be easily found on search engines.

Suggestion: Effect of application of biostimulants on the biomass, nitrate, pigments and antioxidants content in radish and turnip microgreens

Abstract: Explanatory, it presents the significance of the study, the objective, treatments and discusses the results obtained.

Introduction: Very well structured and clarifies the problem of the study.

Lines 50-56: Put paragraphs 5 and 6 in a single paragraph, for addressing the same topic.

Lines 79-85: Put paragraphs 9 and 10 in a single paragraph, for addressing the same topic.

Lines 86-88: Objective is clear and responds to the hypothesis, however, it does not address antioxidant activities, nitrate content and pigments as in the title. The title of all surveys must be the objective response and reflect the conclusion readjust.

Material and Methods: It was well detailed and described the separation into topic improves the extent of assessments, treatments and statistics.

Total phenolics measurement and Radical Scavenging Activity Measurement (references)

Results: The results were presented in a brief and explanatory way. The figures and tables are of excellent quality and are well organized.

PCA and Pearson’s correlations: Is it necessary to keep Pearson's correlations, and principal components? The principal components can accurately clarify the behavior of the two species under the effect of biostimulants in all analyzed variables.

Discussion: The discussion explains the results obtained separately for each species. This brings clarity to the readers, however, the authors did not highlight the gains with the use of the biostimulant as realized in the results. This is very important to support the conclusion.

In the discussion, it is essential to put the percentages of increase of each characteristic when using biostimulants in relation to the control, mainly for having good results that suggest the use of these products in the production of microgreens.

As highlighted in the results and discussion, the use of PCA enhances the quality of the results obtained in each species, I suggest removing the correlations.

Conclusions: In this item, the authors have only 1 conclusion paragraph, so they are not conclusions.

The first sentence of the conclusion is inappropriate for a conclusion, it is an introductory sentence and not needed in the conclusion, be succinct. Or make a paragraph at the end of the discussion with the final considerations with the main results obtained for each species and in the conclusions item just indicate the use of the biostimulant for the species with the best response and the influenced variables.

(The results from the present experiment revealed that the application of biostimulants during growth may be effective on final biomass in microgreens. These results also demonstrated that some nutritional parameters, such as chlorophylls, total phenolic content, antioxidant activity, may be promoted and the nitrate content may be reduced by the application of natural and commercial biostimulants in microgreens of the two species studied (turnip greens and radish).

Redo the conclusion, it should be brief and indicate the use of one or all biostimulants for each species, or indicate the non-use.

Author Response

Dear reviewer,

The authors would like to thank you for your comments. The manuscript has been accordingly revised. Corrections and suggestions have been implemented in the current version of the manuscript. All the modifications are marked with the Track Changes tool in the manuscript. We hereby provide a point-by-point answer.

The authors

Biostimulants application in Brassica rapa L. subsp. oleifera (D.C.) and Raphanus sativus L. microgreens: effects on biomass, nitrate content, pigments, antioxidants and antioxidants activity

The title is too long and leaves the reader confused, because all the study information was included in the title. the title must be explanatory and the species used do not need to be highlighted as scientific name in the title, but popular name (radish and turnip greens) and put the scientific names in the keyword or in the abstract to be easily found on search engines.

Suggestion: Effect of application of biostimulants on the biomass, nitrate, pigments and antioxidants content in radish and turnip microgreens

Author response (A.A.). Thanks for the suggestion. The title and keywords are changed according your suggestion.

Abstract: Explanatory, it presents the significance of the study, the objective, treatments and discusses the results obtained.

Introduction: Very well structured and clarifies the problem of the study.

A.A.: Thanks for the positive comments.

Lines 50-56: Put paragraphs 5 and 6 in a single paragraph, for addressing the same topic.

A.A. Done

Lines 79-85: Put paragraphs 9 and 10 in a single paragraph, for addressing the same topic.

A.A. Done

Lines 86-88: Objective is clear and responds to the hypothesis, however, it does not address antioxidant activities, nitrate content and pigments as in the title. The title of all surveys must be the objective response and reflect the conclusion readjust.

A.A.: More information in the objective were added.

Material and Methods: It was well detailed and described the separation into topic improves the extent of assessments, treatments and statistics.

Total phenolics measurement and Radical Scavenging Activity Measurement (references)

A.A.: The references were added.

Results: The results were presented in a brief and explanatory way. The figures and tables are of excellent quality and are well organized.

PCA and Pearson’s correlations: Is it necessary to keep Pearson's correlations, and principal components? The principal components can accurately clarify the behavior of the two species under the effect of biostimulants in all analyzed variables.

A.A.: In our opinion, Pearson’s correlations may offer a more accurate information about the level of straightness than joins each couple of traits. Anyway, we preferred keeping table for Pearson’s correlations, as supplementary material.

Discussion: The discussion explains the results obtained separately for each species. This brings clarity to the readers, however, the authors did not highlight the gains with the use of the biostimulant as realized in the results. This is very important to support the conclusion.

In the discussion, it is essential to put the percentages of increase of each characteristic when using biostimulants in relation to the control, mainly for having good results that suggest the use of these products in the production of microgreens.

A.A.: According to referee’s suggestion, new information have been included in the Discussion section.

As highlighted in the results and discussion, the use of PCA enhances the quality of the results obtained in each species, I suggest removing the correlations.

A.A.: As above explained, in our opinion, Pearson’s correlations may offer a more accurate information about the level of straightness than joins each couple of traits. Anyway, we preferred keeping table for Pearson’s correlations, as supplementary material.

Conclusions: In this item, the authors have only 1 conclusion paragraph, so they are not conclusions.

A.A.: The conclusion section has been implemented according to referee’s comment.

The first sentence of the conclusion is inappropriate for a conclusion, it is an introductory sentence and not needed in the conclusion, be succinct. Or make a paragraph at the end of the discussion with the final considerations with the main results obtained for each species and in the conclusions item just indicate the use of the biostimulant for the species with the best response and the influenced variables.

(The results from the present experiment revealed that the application of biostimulants during growth may be effective on final biomass in microgreens. These results also demonstrated that some nutritional parameters, such as chlorophylls, total phenolic content, antioxidant activity, may be promoted and the nitrate content may be reduced by the application of natural and commercial biostimulants in microgreens of the two species studied (turnip greens and radish).

Redo the conclusion, it should be brief and indicate the use of one or all biostimulants for each species, or indicate the non-use.

A.A.: The conclusion section has been implemented according to referee’s comment.

Reviewer 2 Report

The title is too long. Try to summarize the title.

The title of the figures and tables have to be independent and self-explanatory. Please try to follow this common rule throughout the manuscript.

The discussion section needs more in-depth evidences from the physiological and biochemical points of view. 

The conclusion section is not enough informative and needs rewriting to address all significant results from the experiment.

Overall, I preferred to have an HPLC analysis of phenolics and flavonoids to decide precisely on the effectiveness of treatments and nutritional quality of the microgreens tested. 

Author Response

Dear reviewer,

The authors would like to thank you for your comments. The manuscript has been accordingly revised. Corrections and suggestions have been implemented in the current version of the manuscript. All the modifications are marked with the Track Changes tool in the manuscript. We hereby provide a point-by-point answer.

The authors

The title is too long. Try to summarize the title.

Author response (A.A.): Thanks for the suggestion. The title was modified in “Effect of application of biostimulants on the biomass, nitrate, pigments and antioxidants content in radish and turnip micro-greens”.

The title of the figures and tables have to be independent and self-explanatory. Please try to follow this common rule throughout the manuscript.

A.A.: According to referee’s suggestion, further information in the figures and tables were added.

The discussion section needs more in-depth evidences from the physiological and biochemical points of view.

A.A.: The ‘Discussion’ section has been implemented with new information. Nevertheless, taking into account the agronomic nature of the present study, as well as of the Journal selected for its potential publication, we didn’t consider any biochemical and physiological aspect, but we only focused on the effects of biostimulants on biomass yield and nutraceutical traits.

The conclusion section is not enough informative and needs rewriting to address all significant results from the experiment.

A.A.: The conclusion section has been implemented according to referee’s comment.

Overall, I preferred to have an HPLC analysis of phenolics and flavonoids to decide precisely on the effectiveness of treatments and nutritional quality of the microgreens tested.

A.A.: The methods used are spectrophometric since the quality evaluation was carried out considering the amount of the compounds considered. We agree with the referee that the HPLC analyses are more accurate, however, for nutritional point of view we focused on total amounts.

Reviewer 3 Report

This manuscript investigated the effects of application of biostimulants on yield and nutraceutical properties of turnip greens and radish microgreens, to verify the beneficial effects of biostimulants on yield and nutraceutical properties of microgreens. This is interesting study.

There are some problems as follow:

1. How many seeds sown in plastic perforated trays?

2. The seed of turnip usually is smaller than radish,why the fresh biomass per unit area of turnip microgreens  was higher than radish (Fig1 (a) vs (b))?

3. Why the seedling height of turnip was two times more than radish (Fig2 (a) vs (b))? And then why the fresh weight of single seedling at harvest was greater in microgreens of radish (207.2 213mg vs. 37.6 mg in turnip greens) (Figure 2c,d)? These were antilogy.

4. Why seedling height of turnip in Bio-1 treatment was significantly lower than other treatments while fresh weight of single seedling was no significantly different(Fig2 (a) vs (c))?

5. The value of the total sugars content in these two species of microgreens is too low(Fig 5).

6. The unit of DPPH radical scavenging is error ( Fig. 7).

7. Why ‘ Chl b is not necessary to the photosynthetic process’ (line 369-370)

Author Response

Dear reviewer,

The authors would like to thank you for your comments. The manuscript has been accordingly revised. Corrections and suggestions have been implemented in the current version of the manuscript. All the modifications are marked with the Track Changes tool in the manuscript. We hereby provide a point-by-point answer.

The authors

There are some problems as follow:

  1. How many seeds sown in plastic perforated trays?

Author response (A.A.). Seeds used per each tray were approx. 3500, for radish, and approx. 8000, for turnip greens. The information have been reported in materials and methods.

  1. The seed of turnip usually is smaller than radish, why the fresh biomass per unit area of turnip microgreens was higher than radish (Fig1 (a) vs (b))?

A.A.: Referee is right. However, a greater number of seeds used for turnip green (see above response to comment n. 1), thus, greater number of seedlings/unit area, may have more than compensated a lower single seedling fresh weight in turnip, leading to a greater final fresh biomass/unit area than radish.

  1. Why the seedling height of turnip was two times more than radish (Fig2 (a) vs (b))? And then why the fresh weight of single seedling at harvest was greater in microgreens of radish (207.2 213mg vs. 37.6 mg in turnip greens) (Figure 2c,d)? These were antilogy.

A.A.: Greater number of seedlings/unit area measured in turnip probably induced a lengthening of hypocotyls due to stronger plant competition, and, then, microgreens which were higher but much thinner. Seedlings in radish, although smaller, were much thicker than those of turnip, and with greater weight.

  1. Why seedling height of turnip in Bio-1 treatment was significantly lower than other treatments while fresh weight of single seedling was no significantly different (Fig2 (a) vs (c))?

A.A.: Referee is right. We may suppose that greater variability in seedling weight within the seedling population occurred, that could justify the lack of significance at ANOVA.

  1. The value of the total sugars content in these two species of microgreens is too low (Fig 5).

A.A.: The referee is right. Nevertheless, we found some literature reporting similar low levels of sugars in microgreens. The paragraph has been revised accordingly and new references have been added.

  1. The unit of DPPH radical scavenging is error (Fig. 7).

A.A.: Unit for DPPH has been changed as mmol TE g-1 FW. Figures and text have been corrected accordingly.

  1. Why ‘Chl b is not necessary to the photosynthetic process’ (line 369-370).

A.A.: We found this statement in literature (WojdyÅ‚o, A.; Nowicka, P.; Tkacz, K.; Turkiewicz, I.P.; Sprouts vs. microgreens as novel functional foods: variation of nutritional and phytochemical profiles and their in vitro bioactive properties. Molecules 2020, 25, 4648; doi:10.3390/molecules25204648). Like also reported by Voitsekhovskaja, O. V., & Tyutereva, E. V. (2015). Chlorophyll b in angiosperms: functions in photosynthesis, signaling and ontogenetic regulation. Journal of Plant Physiology, 189, 51-64. ‘Despite the fact that the differences between molecular structures of Chla and Chlb are tiny, Chlb, with two exceptions (see below), does not occur in photosystems but is present only in the antenna complexes. In nature, Chlb never occurs in photosystems, except for the cyanobacterial group Prochlorophyta (in some Prochlorophyta, ChlÉ‘ and Chlb are replaced by divinyl ChlÉ‘ and divinyl Chlb, respectively) and the unicellular Prasinophycean green alga Micromonas (Ito and Tanaka, 2011; Kunugi et al., 2013)’.

Round 2

Reviewer 3 Report

My comments are correctly responded.